# Pigment Identification and Gene Expression Analysis during Erythrophore Development in Spotted Scat (*Scatophagus argus*) Larvae

**DOI:** 10.3390/ijms242015356

**Published:** 2023-10-19

**Authors:** Yongguan Liao, Hongjuan Shi, Tong Han, Dongneng Jiang, Baoyue Lu, Gang Shi, Chunhua Zhu, Guangli Li

**Affiliations:** 1Guangdong Research Center on Reproductive Control and Breeding Technology of Indigenous Valuable Fish Species, Fisheries College, Guangdong Ocean University, Zhanjiang 524088, China; lyg18621324806@163.com (Y.L.); shihj@gdou.edu.cn (H.S.); ht2405039631@163.com (T.H.); dnjiang@gdou.edu.cn (D.J.); shign@126.com (G.S.); zhuch@gdou.edu.cn (C.Z.); 2School of Life Sciences, Guangzhou University, Guangzhou 510006, China; lby12530@163.com

**Keywords:** spotted scat, erythrophore, pigment, gene expression

## Abstract

Red coloration is considered an economically important trait in some fish species, including spotted scat, a marine aquaculture fish. Erythrophores are gradually covered by melanophores from the embryonic stage. Despite studies of black spot formation and melanophore coloration in the species, little is known about erythrophore development, which is responsible for red coloration. 1-phenyl 2-thiourea (PTU) is a tyrosinase inhibitor commonly used to inhibit melanogenesis and contribute to the visualization of embryonic development. In this study, spotted scat embryos were treated with 0.003% PTU from 0 to 72 h post fertilization (hpf) to inhibit melanin. Erythrophores were clearly observed during the embryonic stage from 14 to 72 hpf, showing an initial increase (14 to 36 hpf), followed by a gradual decrease (36 to 72 hpf). The number and size of erythrophores at 36 hpf were larger than those at 24 and 72 hpf. At 36 hpf, LC–MS and absorbance spectrophotometry revealed that the carotenoid content was eight times higher than the pteridine content, and β-carotene and lutein were the main pigments related to red coloration in spotted scat larvae. Compared with their expression in the normal hatching group, *rlbp1b*, *rbp1.1*, and *rpe65a* related to retinol metabolism and *soat2* and *apoa1* related to steroid hormone biosynthesis and steroid biosynthesis were significantly up-regulated in the PTU group, and *rh2* associated with phototransduction was significantly down-regulated. By qRT-PCR, the expression levels of genes involved in carotenoid metabolism (*scarb1*, *plin6*, *plin2*, *apoda*, *bco1*, and *rep65a*), pteridine synthesis (*gch2*), and chromatophore differentiation (*slc2a15b* and *csf1ra*) were significantly higher at 36 hpf than at 24 hpf and 72 hpf, except for *bco1*. These gene expression profiles were consistent with the developmental changes of erythrophores. These findings provide insights into pigment cell differentiation and gene function in the regulation of red coloration and contribute to selective breeding programs for ornamental aquatic animals.

## 1. Introduction

Fish body color is involved in population communication, sexual selection, and protection against predators [1]. Fish coloration and patterns are determined by the distribution of chromatophores on scales and skin. Red coloration dominated by erythrophores is a valuable economic trait and serves as an honest signal of individual quality in mate choice [2,3]. In teleosts, six types of chromatophores have been identified to date, including melanophores, erythrophores, xanthophores, iridophores, leucophores, and cyanophores [4]. Many reports have identified transcription factors involved in chromatophore fate specification, especially in determining melanophore, xanthophore, and iridophore differentiation in fish, including *wnt* (wingless-type), *csf1ra* (colony-stimulating factor 1 receptor, a), *pax3* (paired box 3), *pax7* (paired box 7), *mitfa* (melanocyte-inducing transcription factor a), *foxd3* (forkhead box D3), and *ltk* (leukocyte receptor tyrosine kinase) [5,6,7,8,9,10]. However, genes determining fate differentiation and cell populations of erythrophores are still largely unexplored.

The complete biosynthetic pathway of melanin and pteridine pigments has been proposed, and these pigments are endogenous substances synthesized autonomously in organs and tissues of teleosts [11,12]. Melanin formation in teleosts is regulated by the melanocortin system, including adrenocorticotropic hormone, α-MSH (α-melanotropic hormone), Asip1 (agouti-signaling protein 1), and melanocortin receptors Mc1r (melanocortin 1 receptor) and Mc5r (melanocortin 5 receptor) [2]. *asip1* and *agrp2* (agouti-related neuropeptide 2) interact with *mc1r* and *mc5r* to regulate the dorsal and ventral melanin pattern in teleosts [13,14,15]. Carotenoids are important pigments with essential roles in animal body coloration, organismal immunity [16,17], mate choice, social competition, and species recognition [18,19]. Bright coloration (red, orange, and yellow) formed by carotenoids improves the market value of aquatic taxa, particularly fish. Carotenoids cannot be synthesized autonomously in the organs and tissues of animals but are derived from the diets in the external environment and stored in bodies. Carotenoids belong to exogenous pigments [20]. Carotenoids are commonly added to bait to ensure the coloration of fish, such as salmon (*Oncorhynchus*) and trout (*Salmo playtcephalus*), and the health of shellfish, such as shrimp (*Penaeus*) and lobster (*Palinuridae*) [21]. However, little is known about the molecular mechanisms underlying carotenoid metabolism and erythrophore differentiation. Carotenoid metabolism in vertebrates can be divided into three main biological processes: carotenoid uptake and transport, binding and deposition, and catabolic processes. Candidate genes involved in these processes have been reported [22]. Of these, two members of the scavenger receptor family, *scarb1* (scavenger receptor type B 1) and *cd36* (decision cluster 36), are involved in carotenoid uptake and transport [23,24,25]. Membrane-associated proteins, such as members of the GSTS family (*gstp1*, glutathione *S*-transferase pi 1) [20,26], star family member (*stard3*, StAR-related lipid transfer domain-containing 3) [27,28], perilipin family (*plin2*, perilipin 2) [29], and lipocalin protein family (*apod*) [20,30], are involved in carotenoid binding and deposition. In carotenoid catabolism, β-carotene-15’,15’-monooxygenase (encoded by *bco1*) and β-carotene 9’,10’-carotenoid oxygenases (encoded by *bco2*) [30,31,32] are the main carotenoid oxygenases for the degradation of carotenoids. Research has focused on the mechanisms by which carotenoids contribute to coloration. However, this process is still not fully understood.

Spotted scat is a marine aquaculture fish with high economic and ornamental value. The transcriptome is an important technique for studying body color. Black spot formation and melanophore coloration have been evaluated [33]. Red coloration is essential for ornamental value in fish, erythrophore development, and red pigmentation in the species remains to be clarified. In spotted scat, erythrophores are present at the embryonic stage even before hatching, making it an ideal species for studying erythrophore formation and red coloration. However, erythrophores in the skin are gradually covered by melanophores when they are formed. Therefore, in this study, erythrophore formation was observed after the inhibition of melanin by PTU, and the main pigments related to erythrophore coloration were identified by liquid chromatography/mass spectrometry (LC–MS) and spectrophotometry. In addition, the expression patterns of genes related to red coloration and the effect of PTU on pigmentation in spotted scat larvae were explored by transcriptome and quantitative real-time PCR (qRT-PCR) analyses. The potential genes involved in erythrophore formation and carotenoid metabolism in spotted scat were revealed, providing new insights into the red coloration of aquatic animals. The results will facilitate an understanding of the molecular mechanisms of erythrophore differentiation and red coloration in other marine fish species and may benefit the selective breeding programs for ornamental and cultured fish.

## 2. Results

### 2.1. Microscopic Observation of Erythrophores in Spotted Scat during Hatching

To evaluate erythrophore development, embryos were treated with PTU (0.003%) to inhibit the formation of melanin. Embryos in the control group were incubated in natural seawater without PTU. Before the embryos hatched (14 hpf), microscopic observation showed that melanophores were distributed in the whole trunk and yolk sac of embryos in the control group but were not detected in the treatment group, and obvious erythrophores were visible in the caudal fold membrane (Figure 1a,b). From 16 to 36 hpf, more erythrophores appeared and were distributed on the eyes, notochord, and, most prominently, caudal myomeres of the trunk in the treatment group. However, they were covered by melanophores in the control group (Figure 1c–j), and few erythrophores were observed at the edge of the yolk sac at 36 hpf (Figure 1j). At 48 and 72 hpf, erythrophores decreased in the trunk and assembled in the venter of spotted scat larvae in the treatment group, while they were undetectable and covered by melanophores in the control group (Figure 1k–p).

### 2.2. Illumina Sequencing and Annotation, Functional Enrichment Analyses of Differentially Expressed Genes (DEGs)

Based on the inhibition of melanin synthesis and melanophore differentiation by PTU and the obvious increase in melanophores and erythrophores at 36 hpf, larvae in the control and PTU treatment groups were chosen for transcriptomics and gene expression analysis. After quality control of sequencing data, 124.30 million clean reads and 37.08 billion clean bases were obtained. The GC content and Q30 values exceeded 48% and 95%, respectively (Table 1). 

A principal component analysis (PCA) was conducted to explore the consistency in gene expression profiles between the control and PTU treatment groups. All the biological replicates were aggregated, and different treatments were clearly separated (Figure 2A), where 76.3% and 16.4% of the total variance was explained by the first and second principal components (PCs), respectively. Transcriptome analysis revealed 43 DEGs between the control and PTU treatment groups, of which 32 genes were up-regulated, and 11 genes were down-regulated in the PTU treatment group (Figure 2B). 

A GO enrichment analysis showed that DEGs between groups were enriched for terms in the three main branches (Padj < 0.05): Biological Process (BP), Molecular Function (MF), and Cellular Component (CC). For BP, the majority of the up-regulated genes were associated with glycerolipid metabolic process (GO:0046486), cellular amide metabolic process (GO:0043603), and down-regulated genes were associated with positive regulation of cell projection organization (GO:0031346). Enriched terms in the CC category mainly included cytoplasmic vesicle lumen (GO:0060205), postsynapse (GO:0098794) for up-regulated genes, and synaptic membrane (GO:0097060) for down-regulated genes. The significant terms in the MF category were triglyceride lipase activity (GO:0004806), dicarboxylic acid transmembrane transporter activity (GO:0005310), and G-protein-coupled receptor activity (GO:0004930) (Figure 2C,D).

KEGG pathway analysis showed that DEGs were significantly involved in phototransduction (ko04744), retinol metabolism (ko00830), linoleic acid metabolism (ko00591), metabolism of xenobiotics by cytochrome P450 (ko00980), steroid hormone biosynthesis (ko00140), inositol phosphate metabolism (ko00562), and steroid biosynthesis (ko00100). The figure below displays the top 20 enriched pathways (with the smallest q-value) (Figure 3).

### 2.3. Validation of RNA-Seq Data with qRT-PCR

To explore the physiological effects of PTU at the genetic level, six genes were selected for validation using qRT-PCR. Each group included three biological replicates. Of these, *rlbp1b*, *rbp1.1*, and *rpe65a* participated in the retinol metabolism pathway, and *soat2* and *apoa1* were related to steroid hormone biosynthesis and steroid biosynthesis. In addition, *rh2* participated in phototransduction (Table 2). By qRT-PCR, the PTU treatment group showed higher mRNA expression levels of *rlbp1b* (*p* < 0.01), *rbp1.1* (*p* < 0.01), *soat2* (*p* < 0.01), *apoa1* (*p* < 0.01), and *rpe65a* (*p* < 0.01) and lower levels of *rh2* (*p* < 0.01) than the control group (Figure 4), consistent with the transcriptome results. These findings indicated that the transcriptome sequencing results were reliable.

### 2.4. Characteristics and Pigments Responsible for Red Coloration in Spotted Scat Larvae

Considering the observation of erythrophore changed in spotted scat larvae of the PTU treatment group (Figure 1), larvae at 24, 36, and 72 hpf were chosen for the subsequent characteristics, pigments, and gene expression analysis for red coloration and erythrophore development. Carotenoid droplets of erythrophores were visible from 24 to 72 hpf, with larger droplets observed at 36 hpf (Figure 5A). The number and diameter of erythrophores at 36 hpf were significantly larger than those at 24 hpf and 72 hpf (Figure 5B,C). Larvae at 36 hpf were chosen for subsequent pigment analyses. By LC–MS, two types of carotenoids, β-carotene and lutein, were identified for red coloration, and β-carotene was the most abundant. Astaxanthin, fucoxanthin, and l-sepiapterin were not detected at 36 hpf (Figure 5D and Appendix A). Spectrophotometry revealed that carotenoids (4.32 mg/kg) were significantly more abundant than pteridine pigments (0.511 mg/kg) in the spotted scat larvae at 36 hpf (Figure 5E).

### 2.5. Expression of Genes Responsible for Red Coloration in Spotted Scat Larvae

The expression levels of nine genes (Table 3) involved in carotenoid metabolism (*scarb1*, *plin6*, *plin2*, *apoda*, *bco1*, *rep65a*), pteridine synthesis (*gch2*), and chromatophore differentiation (*slc2a15b* and *csf1ra*) were measured by qRT-PCR. Expression levels of these genes were significantly higher at 36 hpf than at 24 and 72 hpf (*p* < 0.05), except for *bco1* (*p* < 0.05), which showed no difference in expression between 24 and 72 hpf (Figure 6).

## 3. Discussion

Red coloration is considered an economically important trait in ornamental fish species. The mechanism underlying red coloration is not well understood. The synthesis, transport, and deposition of pigment substances on the epidermis are decisive processes for the formation of skin color phenotypes [22]. To characterize the development of erythrophore and red coloration in spotted scat, inhibitor treatment, transcriptome analysis, LC–MS, and spectrophotometry were used. 

### 3.1. Effects of PTU on Melanin Pigment Formation and Gene Expression

PTU can inhibit tyrosine hydroxylase activity and reduce the conversion from tyrosine to melanin [34]. Therefore, PTU is used to inhibit melanin formation in embryos and increase optical transparency for embryological observation in studies of various taxa, such as ascidian larvae (*Styela partita* and *Ciona intestinalis*) [34], tadpole (*Rana pipiens*) [35], zebrafish (*Danio rerio*) [36], and mangrove killifish (*Kryptolebias marmoratus*) [37]. Several physiological effects of PTU on zebrafish have been reported, including alterations in hatching and survival rates, eye size, and visual behavior [36,38,39]. 

In this study, spotted scat embryos were treated with 0.003% PTU from 0 to 72 hpf. Melanin disappeared in embryos of the PTU treatment group. However, expression levels of some well-known genes related to melanin formation, such as tyrosinase (*tyr*) and tyrosinase-related protein 1 (*tyrb1*), were not altered significantly. In addition to inhibiting melanin synthesis, PTU alters substance transport and other metabolic pathways that indirectly influence melanogenesis. *soat2* and *apoa1*, involved in steroid biosynthesis, were significantly up-regulated after PTU treatment. In zebrafish, *soat2* has catalytic activity in the synthesis of cholesterol esters and participates in the transport of yolk cholesterol during zebrafish embryonic development [40]. The *apoa1* gene functions together with high-density lipoprotein to transport carotenoids and scavenge free radicals [41]. *apoa1* participates in steroid biosynthesis [42,43]. In zebrafish embryos, autophagic activation and lysosomal accumulation were observed after treatment with 0.003% PTU [44]. Larvae might regulate energy metabolism and stress responses via steroid hormone biosynthesis in response to the adverse effects of PTU. Moreover, PTU delays the development of the inner segment of photoreceptor cells and enlarges intercellular space between the retinal inner and outer leaflets in Burton’s mouthbrooder (*Astatotilapia burtoni*) [45]. RH2 is an opsin protein in vertebrates and is expressed in the photoreceptor cells of the retina, participating in phototransduction [46,47]. In this study, PTU decreased the expression level of *rh2* and affected the phototransduction pathway in spotted scat larvae. It might inhibit the formation of melanin in the pigment layer of the retinal epithelium, slowing the development of the photoreceptor inner segment.

Retinol is ingested from food and transported by retinol-binding protein (RBP) through the bloodstream to various tissues. Retinol is converted into retinol esters and utilized in the blood and liver [48]. This process is accomplished by transporting retinol and retinal via *rbp1* and retinaldehyde-binding protein 1b (*rlbp1b*) [49]. In zebrafish, the knockout of *rlbp1b* impairs retinol metabolism, causing subretinal lipid deposits and photoreceptor degeneration [50]. It promotes the *rep65a*-mediated isomer hydrolase reaction and oxidation of 11cis-retinol to 11cis-retinal [51]. *rbp1* is regarded as an intracellular regulator of vitamin A metabolism and retinol transport, altering African clawed frog (*Xenopus laevis*) anterior neural development [52]. In this study, *rep65a*, *rlbp1b*, and *rbp1* were significantly up-regulated in the PTU treatment group, and the retinol metabolism pathway was affected by PTU treatment. PTU may alter lipid metabolism, photoreceptor development, and neural development via retinol metabolism in spotted scat, but these phenotypes require further study. Retinoic acid signaling promotes the proliferation of melanophores in Japanese flounder (*Paralichthys olivaceus*) [53]. Thus, PTU-induced inhibition of melanin synthesis may be associated with retinoic acid signaling in spotted scat. In addition, 0.003% PTU altered retinoic acid and insulin-like growth factor (IGF) regulation of neural crest and mesodermal components of craniofacial development in zebrafish [54]. However, the mechanism by which PTU affects the synthesis of retinoic acid is unknown. The results of this study suggest that PTU influences retinoic acid signaling via retinol metabolism.

### 3.2. Characteristics, Pigment Identification, and Expression of Genes Responsible for Red Coloration

In this study, erythrophores were detected in the caudal fin fold membrane at 14 hpf before embryos hatched and were observed in the eyes and notochord; in particular, the caudal part of the trunk from 16 to 36 hpf, with a gradual decrease and assembly in the venter of spotted scat larvae from 48 to 72 hpf. The erythrophore development pattern in spotted scat was similar to that of Maldivian clownfish (*Amphiprion nigripes*) [55]. In some fish, red coloration is dominated by carotenoids such as astaxanthin and β-carotene [56]. Fucoxanthin, astaxanthin, lutein, and β-carotene have been detected in the muscle, liver, and gonad (testis and ovary) of adult spotted scat [57]. In this study, carotenoid and pteridine pigments were identified and measured by spectrophotometry and LC–MS at 36 hpf. The content of carotenoids was about eight times higher than that of pteridine, while the astaxanthin, fucoxanthin, and l-sepiapterin pigments were not detected. These results suggested that carotenoids were the predominant pigment mediating the red coloration of erythrophores in spotted scat. However, the pteridine content was relatively low, which is consistent with observations in Oujiang color common carp *(Cyprinus carpio* var. color) [56] and Trinidad guppies (*Poecilia reticulata*) [58]. In birds [59,60,61], such as weaverbirds (*Ploceidae*) and some reptiles, including lizards, red coloration can be caused by carotenoids [62,63]. In fish, carotenoids cannot be synthesized but are obtained and absorbed from feed and then transported from the intestine to tissues, such as muscle or skin, for deposition of coloration [64]. However, before and during hatching in spotted scat, the digestive tube is not well developed, limiting the ability to feed from the external environment. The carotenoid droplets in erythrophores can contribute to the maternal carotenoid pigments. Maternal pigments are essential for egg protection, hatching, and immunity [65,66,67]. However, the molecular mechanism underlying maternal pigmentation during embryonic and larval stages remains unclear.

Unlike melanin and pteridine, our understanding of the molecular genetics of carotenoid coloration in fish is limited [68]. Within this limitation, considering the dietary origin of carotenoids, we focused on their uptake, binding, transport, and deposition by investigating the expression of related genes. Scavenger receptor class B, member 1 (*scarb1*), is necessary for carotenoid uptake and transport [69,70,71]. In Oujiang color common carp, *scarb1* is involved in carotenoid coloration of the skin, and the knockout of *scarb1* resulted in discoloration [56]. In addition, *plin2* expression in queen loach (*Botia dario*) was higher in skin with high redness values [72,73]. In this study, the expression of *scarb1* and *plin2* was significantly higher at 36 hpf than at 24 and 72 hpf, consistent with the changes in the number and carotenoid droplet diameters, consistent with the activation of carotenoid transport for skin coloration in spotted scat. *plin6* could bind to the surface of pigments containing carotenoid droplets, and the knockout of *plin6* in zebrafish caused a significant reduction in β-carotene accumulation [29]. In this study, *plin6* expression was higher when the number and diameter of carotenoid droplets were larger at 36 hpf, suggesting an increase in carotenoid deposition, than at 24 and 72 hpf. For carotenoid catabolism, Bco1 is a classical carotenoid oxygenase that cleaves carbon–carbon double bonds and combines two oxygen atoms in the substrate at the cleavage site [74]. The *bco1* mRNA expression was lower in spotted scat at 36 hpf than at 24 and 72 hpf. These gene expression profiles indicated that the activation of carotenoid transport and deposition and the inhibition of catabolism are responsible for red coloration.

In addition to carotenoids, pteridines are important pigments for xanthophore and erythrophore coloration. GTP cyclization hydrolase, encoded by *gch2*, is the rate-limiting enzyme of the pteridine metabolic pathway [12]. In this study, pteridine pigments were measured by spectrophotometry. Together with the increases in the number of erythrophores and cell diameters at 36 hpf, the expression of *gch2* also increased. In addition, *gch2* expression was significantly higher in red skin than in gray skin in red crucian carp (*Carassius auratus*, red var.) [75]. In zebrafish, *gch2* was necessary for the maintenance of xanthophores, and the knockout of the *gch2* gene resulted in a significant reduction in xanthophores [76]. Thus, the *gch2* mRNA expression level was closely related to xanthophore and erythrophore coloration. Xanthophores and erythrophores are differentiated from the same precursor cells in pearl danio (*Danio albolineatus*) [71]. In zebrafish, *slc2a15b* was highly expressed in xanthophores and neural crest cells [77] and is an important candidate gene for xanthophore lineage differentiation [78]. *csf1ra* is an essential factor for maintaining the xanthophore lineage [79,80]. In this study, *slc2a15b* and *csf1ra* were highly expressed in spotted scat larvae at 36 hpf, together with high cell counts and large diameters. 

## 4. Materials and Methods

### 4.1. Ethics Statement

The experiment was approved by the Institutional Animal Care and Use Committee (IA-CUS) of Guangdong Ocean University and complied with China’s laws and regulations on biological research.

### 4.2. Experimental Design, Fish Treatment, and Sample Collection

All experimental spotted scat embryos were hatched and raised in tanks at 29 °C in Donghai Island, Zhanjiang, Guangdong Province. 

Experiment 1: After fertilization, all embryos were randomly divided into two groups, a control group and a PTU-treated group, with three replicates per group and 200 embryos per replicate. In the PTU-treated group, embryos in tanks were treated with PTU (P7629, ≥98%, Sigma-Aldrich, St. Louis, MO, USA) for 72 h at a standard concentration of 0.003% [36]. PTU was dissolved in natural seawater, and the water was changed once a day (every 24 h). Embryos in the control group were hatched in natural seawater. During hatching, three embryos (biological replicates) were randomly selected and sampled from control and PTU-treated groups at 14, 16, 20, 24, 36, 48, 60, and 72 h post fertilization (hpf). Images were obtained using a ZEISS microscope and ZEN 3.3 (CARL ZEISS., Ltd., Shanghai, China). At 36, 48, and 72 hpf, 15 larvae for each replicate from both PTU-treated and control groups were sampled and stored in an RNA stabilization reagent (Accurate Biology, Changsha, China) for RNA extraction. 

Experiment 2: After fertilization, all embryos were hatched in normal seawater. At 36 hpf, 1 g of larvae (three biological replicates) was randomly selected, added to a 50 mL centrifuge tube, and stored at −80 °C for further pigment identification and experiments.

### 4.3. Reagents and Preparation of Standard Stock Solutions

Astaxanthin standards (CAS No. 472-61-7, 98.1%), LC–MS-grade formic acid (CAS No. 64-18-6, ≥96%), methanol (CAS No. 67-56-1, ≥99.9%), and LC–MS-grade water were purchased from (TMRM) Tan-Mo Technology Corporation (Changzhou, China). Lutein (CAS No. 127-40-2, 92.6%), fucoxanthin (CAS No. 7176-02-5, 99%), β-carotene (CAS No. 7235-40-7, 99.9%), l-sepiapterin (CAS No. 17094-01-8, 95%), and LC–MS-grade acetonitrile (CAS No. 75-05-8, ≥99.9%) were purchased from ANPEL Laboratory Technologies Inc. (Shanghai, China). Analytical reagent petroleum ether (CAS No. 8032-32-4) and HPLC-grade hexane (CAS No. 110-54-3, ≥95%) were purchased from Sinopharm Chemical Reagent Co., Ltd. (Shanghai, China). A nylon filter membrane with a 0.22 µm pore size was purchased from Tianjin JINTENG Experimental Equipment. Co., Ltd. (Tianjin, China). The l-sepiapterin standard was dissolved in methanol, and standards for other pigments were dissolved in acetonitrile. The gradient concentrations of all standard working solutions were 31.25, 62.5, 125, 250, 500, and 1000 ng/mL. Methyl tert-butyl ether (MTBE) (CAS No. 1634-04-4) and NH_4_OH (CAS No. 1336-21-6) were purchased from Chengdu Chron Chemicals Co., Ltd. (Chengdu, China).

### 4.4. Pigment Extraction and Saponification Procedure

For pigment extraction, larvae (0.5 g) were collected into a 50 mL centrifuge tube, 10 mL of petroleum ether was added to the samples, and samples were subjected to shaking for 2 min. Samples were then sonicated for 10 min at 25 °C. The mixture was centrifuged at 12,000 rpm for 3 min to remove the deposition. The pigments were extracted adequately by repeating the above experimental steps 2–3 times. In order to completely remove the residual liquid, the supernatants were transferred into a nitrogen-blowing tube (10 mL) and dried under a stream of nitrogen. Then, 1.0 mL of acetonitrile was added to dissolve the residue. To remove the ester, 3 mL of acetonitrile-saturated hexane (hexane: acetonitrile 8:1 (vol.:vol.)) was added. The mixture was centrifuged at 12,000 rpm for 3 min to collect the supernatant and filtered through a 0.22 μm membrane [56].

### 4.5. Identification of Pigments Using LC–MS

For LC–MS analysis, the extract was separated using the Agilent Poroshell 120 EC-C8 column (100 mm × 2.1 mm × 1.7 µm) (Agilent, Santa Clara, CA, USA) at 30 °C and the Agilent 1290 Ultrahigh-Performance Liquid Chromatography (UHPLC) system (Agilent). The elution method was isocratic, and the mobile phase was LC–MS-grade solvent (acetonitrile: 0.2% formic (volume ratio) 90:10 (vol.:vol.)) with a flow rate of 0.3 mL/min. The injection volume was set to 3 µL.

Mass spectrometry was performed using an Agilent 6470 Mass Spectrometer. Multiple reaction monitoring (MRM) mode was used to confirm the presence of analytes [81], while the ion source and vacuum parameters of the mass spectrometer were optimized to be applicable to all analytes. The ionization mode was electrospray ionization (ESI), and the eluent was monitored by ESI in positive mode. ESI was conducted using a spray voltage of 3.5 kV. High-purity nitrogen gas was used as dry gas at a sheath gas flow rate of 11 L/min. The capillary temperature was set at 350 °C, while the nebulizer pressure was set at 45 psi. The reference values for MRM ion pairs, fragmentation voltage, and collision energy are shown in Table 4. MS data were analyzed using Agilent MassHunter.

### 4.6. Spectrophotometry

The total contents of carotenoid and pteridine pigments in larvae at 36 hpf were measured using absorbance spectrophotometry [82]. Each sample (1 g) was collected into a 5 mL centrifuge tube, and 2 mL of MTBE was added, followed by shaking for 2 min and centrifugation at 3000 rpm for 5 min at 4 °C. The extract was transferred to a new 5 mL centrifuge tube for carotenoid pigment analysis. Subsequently, 2 mL of 1% NH_4_OH solution was added to the extraction residue and shaken for 1 min, followed by centrifugation at 3000 rpm for 5 min at 4 °C. The extract was transferred to a new 5 mL centrifuge tube for pteridine pigment analyses. Absorbance was measured in an ultraviolet spectrophotometer (UV-2600A, Unico Instrument Co., Ltd., Shanghai, China). MTBE was used as a blank control for the carotenoid assay, and a 1% NH_4_OH solution was used as a blank control for the pteridine assay. The contents of carotenoids and pteridines were determined at absorption wavelengths of 447 and 490 nm, respectively [83]. The pigment content was calculated according to the following formula:$$\;X=\frac{A\times K\times V}{E\times M}$$
where X represents the carotenoid or pteridine pigment content (mg/kg), A represents the absorbance value, K represents the constant 10,000, V represents the volume of fixation (mL), E represents the molar extinction coefficient, and M represents the weight of the sample (g). The extinction coefficients for calculating the content of carotenoids and pteridines were 2550 and 10,000, respectively.

### 4.7. Total RNA Extraction, Library Construction, and Sequencing

Total RNA (three tubes per group) from the PTU-treated and control groups was extracted using TRIzol reagent (Life Technologies, Carlsbad, CA, USA) according to the manufacturer’s instructions. The quality of the RNA samples was detected using a NanoDrop2000 Spectrophotometer (Thermo Fisher Scientific, Santa Clara, CA, USA) and by 1% agarose gel electrophoresis. Total RNAs with an RNA integrity number (RIN) score > 7 were selected for sequencing. Six sequencing libraries were constructed and generated by using the NEBNext Ultra RNA Library Prep Kit for Illumina^®^ (NEB, Ipswich, MA, USA) based on the manufacturer’s instructions. After treating the samples with DNase I (NEB), messenger RNAs (mRNAs) were isolated by Oligo (dT)-attached magnetic beads (Illumina, San Diego, CA, USA) and then fragmented by fragmentation buffer reagent (NEB). First-strand cDNA was synthesized with fragmented mRNA as a template and random hexamers as primers, followed by second-strand synthesis. Double-stranded cDNA was synthesized using short fragments as templates. AMPure XP beads were applied to select fragments within the size range of 300–400 bp. Then, 3 μL of USER Enzyme (NEB) was used to select the size, adaptor-ligated cDNA at 37 °C for 15 min, followed by 5 min at 95 °C. PCR was performed using Phusion High-Fidelity DNA polymerase (NEB), universal PCR primers, and Index (X) Primer. Finally, the PCR products were purified by the AMPure XP system, and the quality and quantity of libraries were assessed using the Agilent Bioanalyzer 2100 System and StepOnePlus Real-Time PCR System (Thermo Fisher Scientific, Santa Clara, CA, USA). The index-coded samples were clustered using the cBot Cluster Generation System with the TruSeq PE Cluster Kit v4-cBotHS (Illumina). All cDNA libraries were sequenced on the Illumina NovaSeq 6000 platform.

### 4.8. Data Filtering, Read Mapping, and Differential Expression Analysis

To obtain clean reads, raw reads were filtered using the high-throughput quality control (HTQC) package. High-quality clean data were obtained by filtering reads and removing adapter sequences, and low-quality reads from the Illumina high-throughput sequencing platform and were saved in FASTQ format. All clean libraries were submitted to the NCBI Sequence Read Archive (SRA) database (Accession No.: PRJNA985608). Based on the spotted scat reference genome sequence (https://ngdc.cncb.ac.cn/search/?dbId=gwh&q=GWHAOSK00000000.1, accessed on 3 September 2022); accession number GWHAOSK00000000.1), HISAT2 [84] was used to map RNA-seq clean reads, and String Tie was applied to assemble the mapped reads. 

Gene functions were annotated against the NCBI nonredundant (NR; https://ftp.ncbi.nih.gov/blast/db/, accessed on 11 September 2022), Swiss-Prot (http://www.uniprot.org/, accessed on 11 September 2022), Kyoto Encyclopedia of Genes and Genomes (KEGG; http://www.genome.jp/kegg/, accessed on 11 September 2022), and Gene Ontology (GO; http://www.geneontology.org/, accessed on 11 September 2022) databases using BLASTx (v. 2.2.26; https://blast.ncbi.nlm.nih.gov/, accessed on 11 September 2022) with an E-value cutoff of 1 × 10^−5^.

Gene expression levels were detected using the fragments per kilobase per million (FPKM) method [85]. The DESeq2R package (version 1.16.3) in R (version 3.6.3) [86] was used to identify DEGs between the control and PTU-treated groups. Criteria for DEGs were set as |log_2_ (Fold Change)| ≥ 1 and FDR < 0.05, where the False Discovery Rate (FDR) refers to the adjusted *p*-value. GO enrichment analysis of DEGs was implemented using the GOseq R packages (version 1.24.0) based on the Wallenius noncentral hyper-geometric distribution [87]. KOBAS (version 3.0) [88] was used to evaluate the enrichment of differentially expressed genes for KEGG pathways. GO terms and KEGG pathways with *p* < 0.05 were considered significantly enriched.

### 4.9. Validation of Transcriptome Data by qRT-PCR

To validate the gene expression profiles from the RNA-seq data, six DEGs (FDR < 0.05 and |log_2_FC| ≥ 1) were selected for qRT-PCR. Primers designed for qRT-PCR using NCBI (https://www.ncbi.nlm.nih.gov/tools/primer-blast/index.cgi?LINK_LOC=BlastHome (accessed on 20 December 2022)) are listed in Appendix A. Reverse transcription was carried out using TransScript^®^ Uni All-in-One First-Strand cDNA Synthesis Super Mix for qRT-PCR (TransGen, Beijing, China). Expression levels of DEGs were quantified by qRT-PCR using PerfectStart^®^ Green qRT-PCR SuperMix (TransGen, Beijing, China) on the Roche LightCycler^®^ 96 System (Roche Diagnostics, Basel, Switzerland). The thermal cycling program included an initial denaturation at 94 °C for 30 s, followed by 40 cycles of 94 °C for 5 s, 60 °C for 15 s, and 72 °C for 10 s. In addition, *β-actin* was used as a reference gene to normalize the expression values. The relative expression was calculated using the 2^−ΔΔCt^ method.

### 4.10. Statistical Analysis

Data were expressed as means ± standard error (SE) (n = 3). Independent samples *t* tests were used to analyze differences with a significance level of *p <* 0.05. Cell number and diameter were analyzed in a quantitative framework using ImageJ (Rawak Software Inc., Stuttgart, Germany), and statistical analyses were performed using GraphPad Prism 9.0 (GraphPad Software Inc., San Diego, CA, USA).

## 5. Conclusions

In this study, we characterized the erythrophore development, identified the pigments, and analyzed the expression of genes responsible for red coloration in spotted scat. Erythrophores in spotted scat were clearly observed during the embryonic stage, CD of erythrophores was clearly visible from 24 hpf, and the number and diameter of cells showed an initial increase (from 14 to 36 hpf), followed by a decrease (from 48 to 72 hpf). The concentration of carotenoids was significantly higher than that of pteridines, and β-carotene and lutein were the main pigments contributing to red coloration in spotted scat larvae before feeding from the external environment. The carotenoid droplets in erythrophores contributed to maternal carotenoid pigments. The expression profiles of genes related to carotenoid metabolism and erythrophore differentiation, including *scarb1*, *apoda*, *plin6*, *plin2*, *rep65a*, *bco1*, *gch2*, *slc2a15b,* and *csf1ra* (Figure 7), were consistent with the developmental changes of erythrophores in spotted scat. These findings provide valuable insights into the regulation of red coloration in aquatic animals and benefit the breeding programs of ornamental fish species. The functions, regulation of erythrophore differentiation, and cell fate of these genes need to be studied and clarified in more fish species.

## Figures and Tables

**Figure 1 ijms-24-15356-f001:**
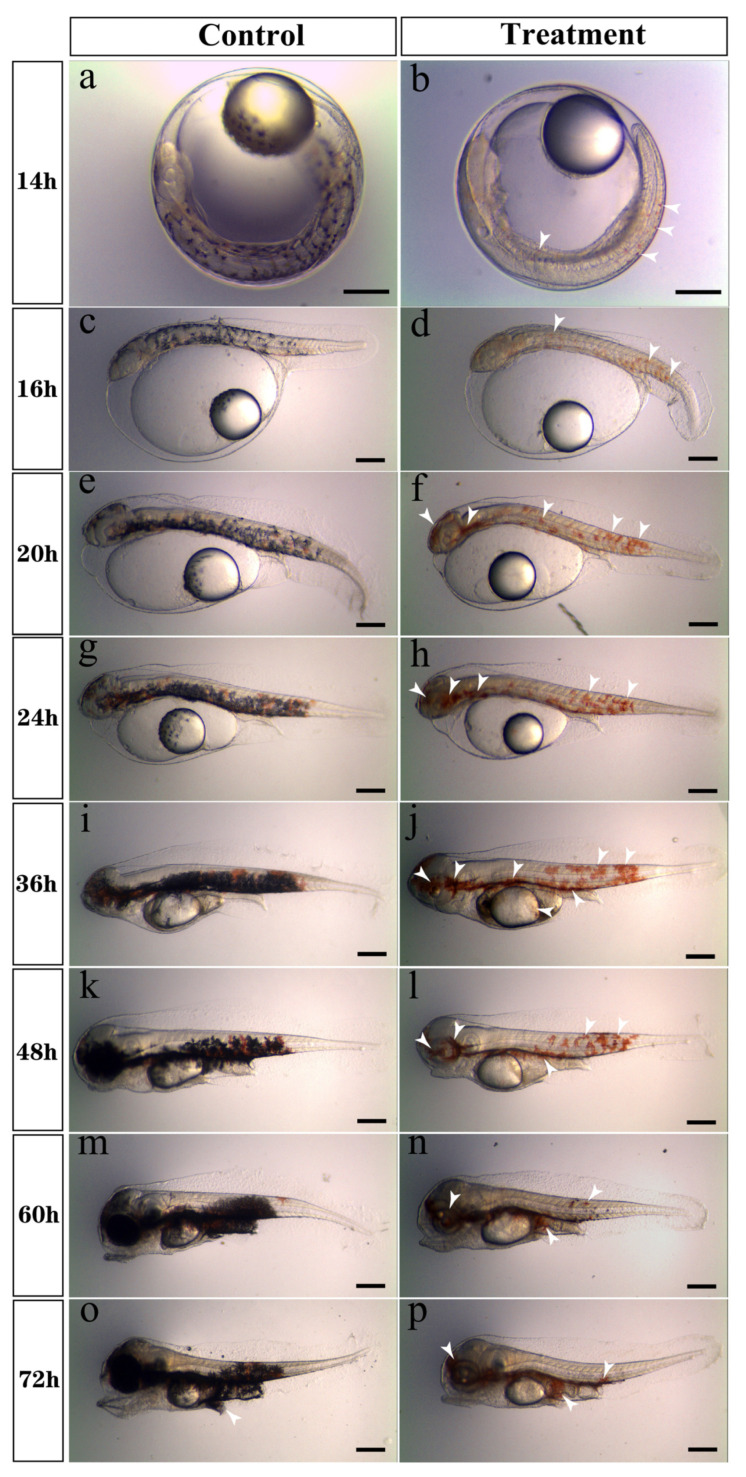
Microscopic observation of erythrophores in spotted scat during hatching. Control, control group; Treatment, PTU-treated group. (**a**,**c**,**e**,**g**,**i**,**k**,**m**,**o**) are the control group from 14–72 h after fertilization, and (**b**,**d**,**f**,**h**,**j**,**l**,**n**,**p**) are the PTU-treated group from 14–72 h after fertilization. Erythrophores are indicated by white arrows (scale bar, 200 μm).

**Figure 2 ijms-24-15356-f002:**
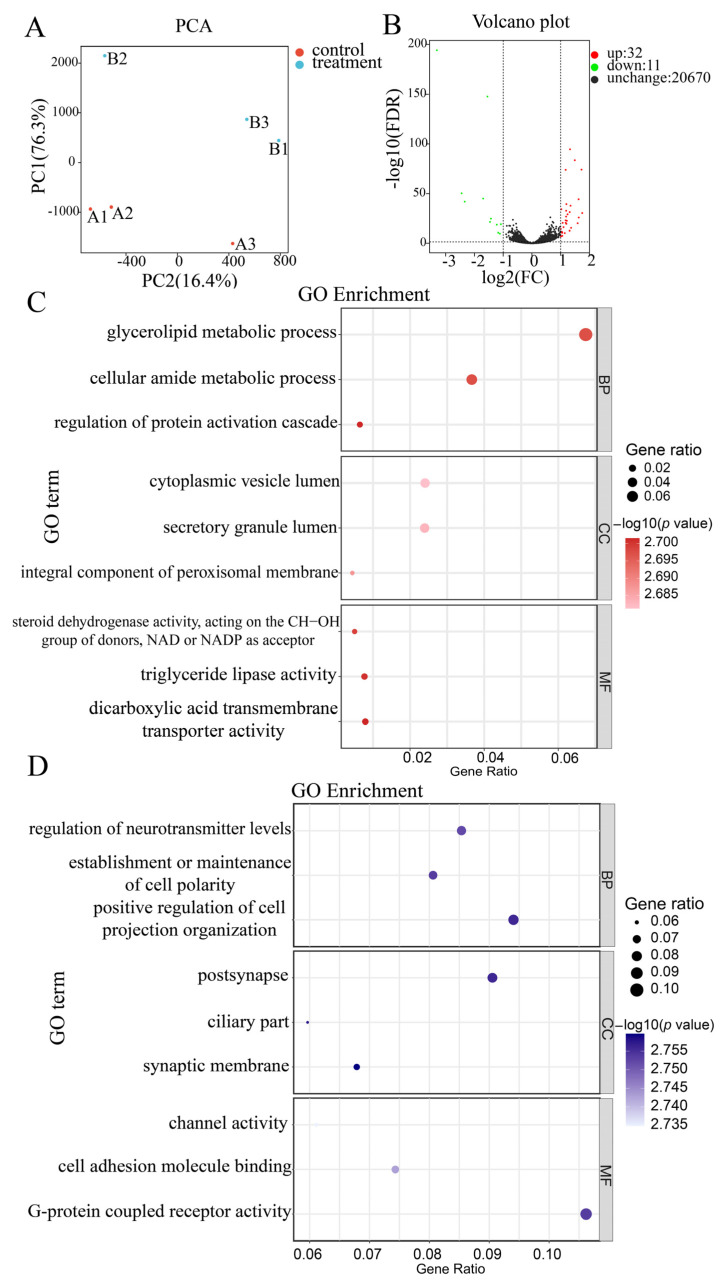
Characterization of DEGs between the control and PTU treatment groups. (**A**) PCA shows the differences between groups. (**B**) In the volcano plot, green dots represent down-regulated genes, red dots represent up-regulated genes, and black dots indicate genes without a significant difference. (**C**,**D**) GO enrichment analysis of DEGs, C, GO terms for up-regulated genes; D, GO terms for down-regulated genes. The circles represent the gene ratio, and the colors represent the *p* value, respectively.

**Figure 3 ijms-24-15356-f003:**
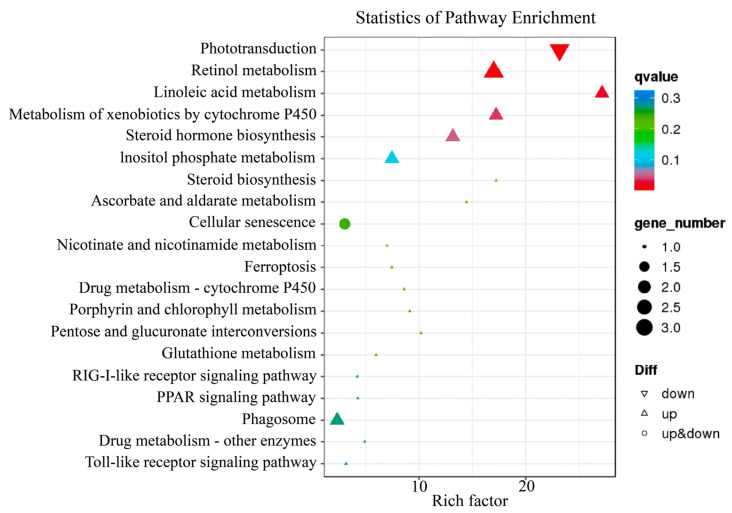
KEGG pathway enrichment analysis of DEGs between the control and PTU treatment groups. The size of the black circle represents the number of DEGs. The colors represent the q-value. The triangle, inverted triangle, and circle represent up-, down-, and up- and down-regulated genes, respectively.

**Figure 4 ijms-24-15356-f004:**
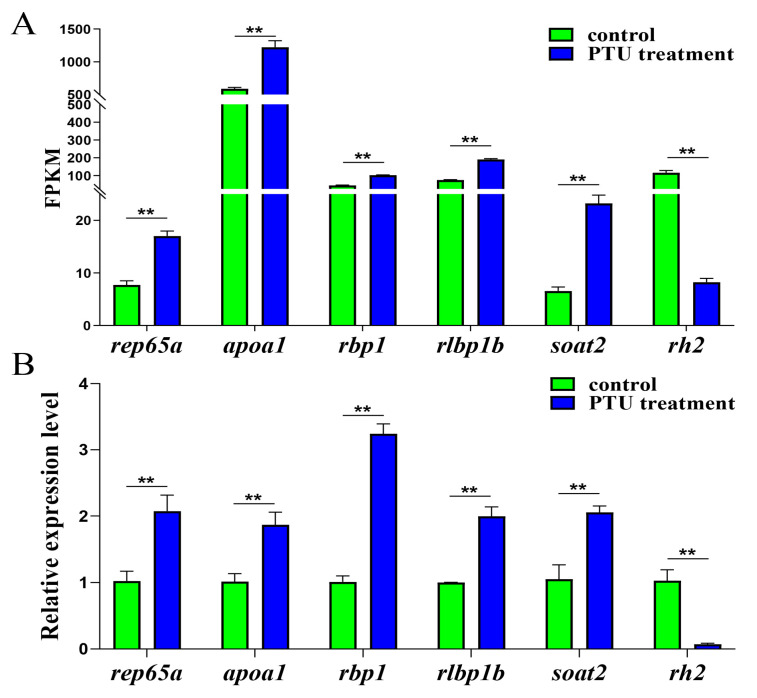
Gene expression validation by RNA-Seq (**A**) and qRT-PCR (**B**). (**A**) FPKM values, (**B**) Relative mRNA expression levels. The green and blue columns represent the control and PTU treatment groups, respectively. The relative expression levels of mRNA transcripts were detected using qRT-PCR via the 2^−∆∆Ct^ method. Data are expressed as means ± standard error (SE) (n = 3). *β*-*actin* was used as the reference gene. The symbols “**” above the error bars indicate significant differences at the levels of *p* = 0.01.

**Figure 5 ijms-24-15356-f005:**
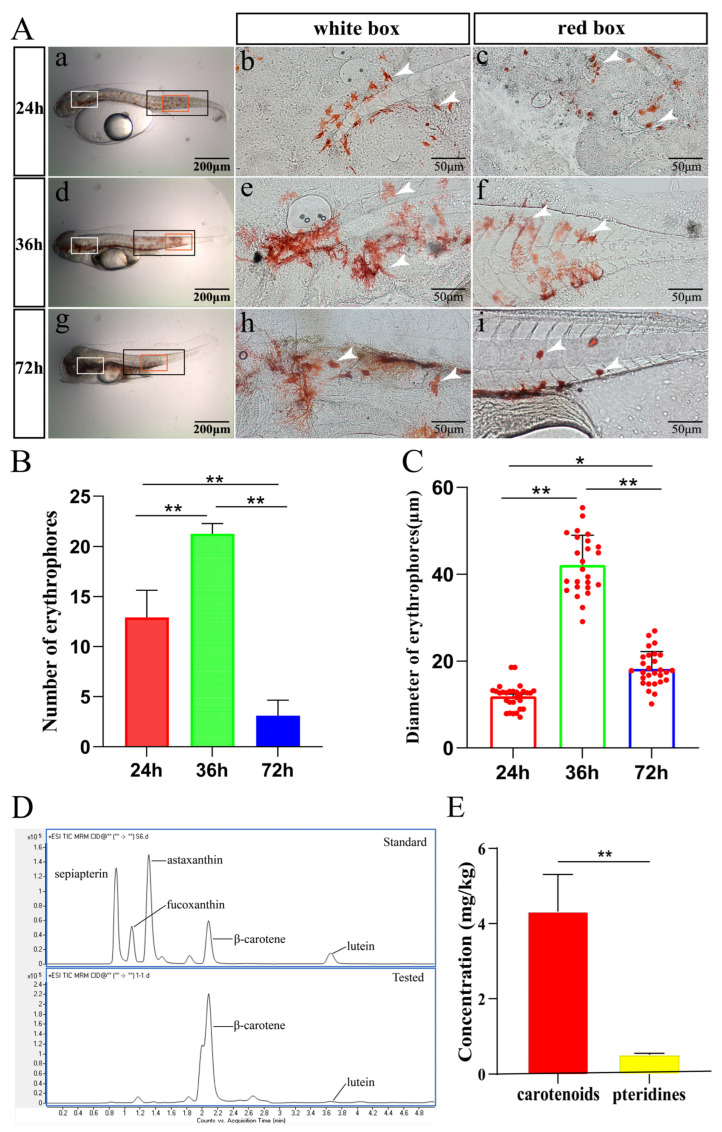
Characteristics (**A**–**C**) and pigment types and contents (**D**,**E**) responsible for red coloration in spotted scat larvae. (**A**) Observation of carotenoid droplets of erythrophores at 24, 36, and 72 h after fertilization. (**b**,**e**,**h**) are the high magnification of erythrophores in the head (white boxed area in (**a**,**d**,**g**)), and (**c**,**f**,**i**) are the high magnification of tail part (red boxed area in (**a**,**d**,**g**)). Erythrophores are indicated by white arrows. (**B**) Numbers of erythrophores at different stages (black box in (**a**,**d**,**g**) of panel (**A**)). (**C**) Diameters of erythrophores at different stages. Red, green, and blue columns represent the data at 24, 36, and 72 h, respectively. (**D**) Identification of pigment types responsible for the red coloration by LC–MS. “Standard” refers to the pigment standards, and “Tested” represents the spotted scat larvae sample. (**E**) Pteridine and carotenoid concentrations at 36 hpf were determined using spectrophotometry (n = 3). The red and yellow columns represent the concentration of carotenoids and pteridines, respectively. Data are presented as means ± SE. The symbols “*” and “**” above the error bars indicate significant differences at the levels of *p* = 0.05 and *p* = 0.01, respectively.

**Figure 6 ijms-24-15356-f006:**
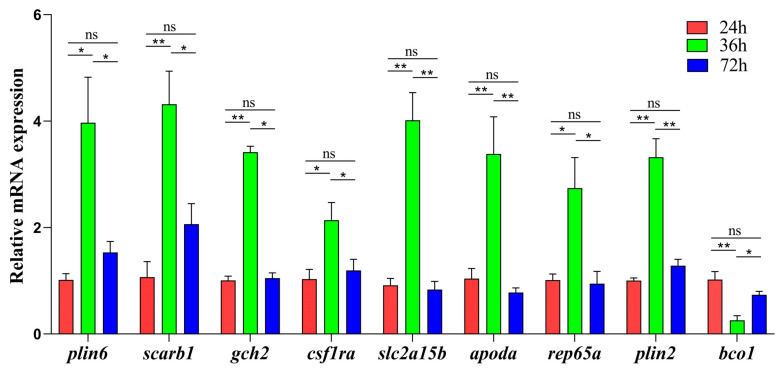
Expression of genes responsible for red coloration in spotted scat larvae at different stages. Red, green, and blue columns represent gene expression at 24, 36, and 72 h, respectively. Relative expression levels were detected using qRT-PCR via the 2^−∆∆Ct^ method. Data are expressed as means ± SE (n = 3). *β*-*actin* was used as the reference gene. The symbols “*”, “**” and “ns” above the error bars indicate significant differences at the levels of *p* = 0.05, *p* = 0.01 and no significant difference, respectively.

**Figure 7 ijms-24-15356-f007:**
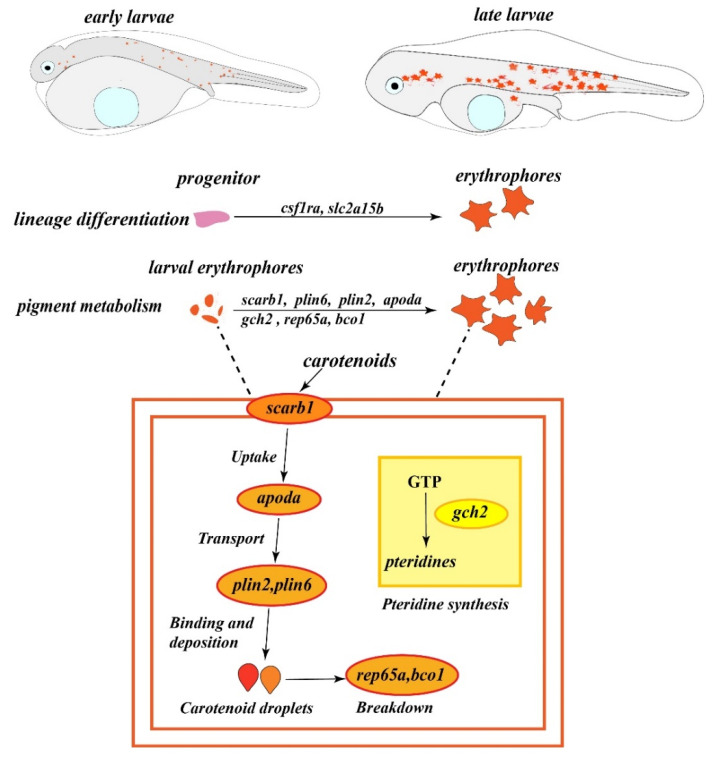
Predictive model for the regulation of erythrophore development and carotenoid metabolism in spotted scat larvae. Genes related to carotenoid metabolism are highlighted by orange circles, and the genes involved in pteridine synthesis are marked by yellow circles.

**Table 1 ijms-24-15356-t001:** Statistics of RNA sequencing data in the control and PTU treatment groups.

Samples	Clean Reads	Clean Bases	GC Content (%)	≥Q30
A1	21,068,282	6,286,267,334	48.78%	95.11%
A2	20,859,439	6,220,965,958	48.84%	95.21%
A3	19,835,579	5,915,579,956	48.76%	95.22%
B1	20,520,674	6,120,618,944	48.76%	95.41%
B2	21,126,146	6,301,031,704	48.96%	95.24%
B3	20,894,446	6,232,585,618	48.71%	95.28%

Note: Samples: sample name; Clean reads: counts of clean PE reads; Clean bases: total length of clean data; GC content: percentage of G and C bases in clean data. ≥Q30 percentage of bases with a Q-score of no less than Q30. A represents the control group, and B represents the PTU treatment group.

**Table 2 ijms-24-15356-t002:** List of DEGs in spotted scat involved in the response to PTU.

Gene ID	Gene Symbol	log2FC	Description
XM_046383247.1	*soat2*	1.625596728	sterol O-acyltransferase 2
XM_046394003.1	*rlbp1b*	1.326384862	retinaldehyde-binding protein 1b
XM_046408526.1	*rbp1*	1.175138631	retinol-binding protein 1
XM_046389465.1	*apoa1*	1.023536727	apolipoprotein A-I-like
XM_046408744.1	*rpe65a*	1.006035995	retinoid isomerohydrolase RPE65 a
XM_046383565.1	*rh2*	−3.307257368	green-sensitive opsin-like

**Table 3 ijms-24-15356-t003:** Genes responsible for red coloration in spotted scat larvae.

Gene ID	Gene Symbol	FPKM	Description
XM_046380961.1	*plin2*	18.11774	perilipin 2
XM_046374291.1	*scarb1*	14.03952	scavenger receptor class B, member 1
XM_046371628.1	*bco1*	4.338945	beta-carotene oxygenase 1
XM_046398563.1	*plin6*	7.46246	perilipin 6
XM_046405845.1	*csf1ra*	3.745435	colony-stimulating factor 1 receptor, a
XM_046376452.1	*gch2*	3.497639	GTP cyclohydrolase 2
XM_046405324.1	*slc2a15b*	3.810482	solute carrier family 2 member 15b
XM_046408801.1	*apoda*	15.3108	apolipoprotein Da, duplicate 1

**Table 4 ijms-24-15356-t004:** Multireaction monitoring ion pair, fragmentation voltage, and collision energy reference values.

Tested Object	Monitoring Ion Pairs (*m*/*z*)	Fragmentation Voltage (V)	Collision Energy (eV)
astaxanthin	598/173.4	100	30
	598/147.3 ^a^	100	25
lutein	570/133.4	120	20
	570/89.2 ^a^	120	30
fucoxanthin	660/642.8	100	5
	660/109.3 ^a^	100	30
β-carotene	538/300.7	120	20
	538/286.2 ^a^	120	30
l-sepiapterin	238/192.1 ^a^	100	10
	238/165.1	100	25

^a^ Quantitative ion pair.

## Data Availability

All clean libraries of RNA sequencing data have been submitted to the Sequence Read Archive (SRA) database (Accession No.: PRJNA985608).

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
