# Peer review of "Pigment Identification and Gene Expression Analysis during Erythrophore Development in Spotted Scat (*Scatophagus argus*) Larvae"

_ijms, 2023, doi:10.3390/ijms242015356_

Round 1
Reviewer 1 Report
This manuscript presents the results of a research work aimed to investigate erythrophore development responsible for red coloration in spotted scat fish embryos.
The results of this work indicate that erythrophores in spotted scat embryos are initially exhibiting an increase, followed by a decrease. Carotenoids, specifically β-carotene and lutein, were identified as the main pigments for red coloration. Gene expression related to carotenoid metabolism and chromatophore differentiation increased at 36 hour post fertilization, corresponding to erythrophore development. This work is interesting and increased our knowledge on red color regulation in fish.
I have several comments and suggestions for improving this work:
1. Introduction. The introduction provides a solid foundation for understanding the importance of fish coloration, the roles of various chromatophores, and the genes associated with coloration in teleosts. However, the introduction could be improved by reducing the frequency of words such as “major” and “mainly”. Furtheremore,given the central focus of the study on erythrophore development and red coloration in spotted scat, the introduction could start by introducing erythrophores and their role in fish coloration. This would provide a more direct support into the specific research objectives. The authos should also clarify why the study on erythrophore development in spotted scat is important. What gaps in knowledge or practical applications make this research relevant? This can help in setting the stage for the reader and emphasizing the significance of the study. The last section of the introduction should be revised to present the objectives or questions that the study aims to address. This will provide readers with a roadmap of what to expect in the subsequent sections of the paper. As the spotted scat is the species of interest, a smooth transition to introducing this species and its characteristics would help set the specific context for the study.
2. Lines 52-54 . The citations style is inconsistent in these lines.
3. Section 2.2: While the initial steps of Illumina sequencing and quality control appear sound, a more comprehensive description of the analysis methods, biological context, and validation steps would enhance the rigor and clarity of the study. The authors provide information on the quality of sequencing data but it would be beneficial to describe the bioinformatics pipeline used for gene annotation and highlight any specific findings or differentially expressed genes between the control and PTU treatment groups. The significance of the study and the relevance of the transcriptome data should be clarified. What specific biological questions or hypotheses are these sequencing data meant to address? How does this fit into the broader context of research on erythrophore development and red coloration in fish? Did the authors perform any pilot study with assays to verify the expression of specific genes or pathways identified through transcriptome analysis?
4. Section 2.4: The approach to validate RNA-seq data with qRT-PCR appears sound, but providing more details on sample size, statistical analysis, and the biological context of gene selection would enhance the overall quality of the validation process. It's essential to mention the sample size (biological replicates) for both the control and PTU treatment groups in the qRT-PCR validation. Additionally, statistical analysis results (e.g., p-values) should be provided to assess the significance of the observed gene expression differences. The authors have chosen specific genes for validation based on their involvement in relevant pathways. However, it's helpful to briefly explain why these particular genes were selected and how their expression changes relate to the research's biological context. This would provide more insight into the significance of these findings. Including some methodological details, such as primer sequences used for qRT-PCR and the cycling conditions, can be valuable for readers who may want to replicate the experiments
5. Section 2.5. There is a discrepancy in the sampling stages used in these two figures. Figure 1 documents erythrophore development at time points ranging from 12 to 60 hours after fertilization (hpf), while Figure 5 presents observations at 24, 36, and 72 hpf. The authors should address this inconsistency and provide a rationale for not using the same stages of development in both figures. It is essential to clarify whether the stages chosen for each figure were deliberate and why they were chosen as such. This will ensure the accuracy and reliability of the comparative analysis between the two figures.
Author Response
This manuscript presents the results of a research work aimed to investigate erythrophore development responsible for red coloration in spotted scat fish embryos.
The results of this work indicate that erythrophores in spotted scat embryos are initially exhibiting an increase, followed by a decrease. Carotenoids, specifically β-carotene and lutein, were identified as the main pigments for red coloration. Gene expression related to carotenoid metabolism and chromatophore differentiation increased at 36 hour post fertilization, corresponding to erythrophore development. This work is interesting and increased our knowledge on red color regulation in fish.
Reply:
Thank you for your positive comments on our work.
I have several comments and suggestions for improving this work:
- Introduction. The introduction provides a solid foundation for understanding the importance of fish coloration, the roles of various chromatophores, and the genes associated with coloration in teleosts. However, the introduction could be improved by reducing the frequency of words such as “major” and “mainly”. Furtheremore,given the central focus of the study on erythrophore development and red coloration in spotted scat, the introduction could start by introducing erythrophores and their role in fish coloration. This would provide a more direct support into the specific research objectives. The authors should also clarify why the study on erythrophore development in spotted scat is important. What gaps in knowledge or practical applications make this research relevant? This can help in setting the stage for the reader and emphasizing the significance of the study. The last section of the introduction should be revised to present the objectives or questions that the study aims to address. This will provide readers with a roadmap of what to expect in the subsequent sections of the paper. As the spotted scat is the species of interest, a smooth transition to introducing this species and its characteristics would help set the specific context for the study.
Reply:
Thank you for your suggestion, the suggestion was adopted. We have reduced words such as “major” and “mainly” in the introduction. We have added more introduction about erythrophores and their importance for spotted scat red coloration. Meanwhile, based on your comments, we have presented questions that the study aims to address in the revised manuscript.
Lines 41-43
Red coloration dominated by erythrophores is a valuable economic trait and serves as an honest signal of individual quality in mate choice [2,3].
Lines 84-90
The transcriptome is an important technique for studying body color. Black spot formation and melanophore coloration have been evaluated[33]. Red coloration is essential for ornamental value in fish, erythrophore development, and red pigmentation in the species remains to be clarified. In spotted scat, erythrophores are present at the embryonic stage even before hatching, making it an ideal species for studying erythrophore formation and red coloration. However, erythrophores in the skin are gradually covered by melanophores when formed.
- Lines 52-54. The citations style is inconsistent in these lines.
Reply:
Thank you for your suggestion, the suggestion was adopted. We have corrected the citations style in the revised manuscript.
[13–15]
- Kobayashi, Y.; Hamamoto, A.; Takahashi, A.; Saito, Y. Dimerization of melanocortin receptor 1 (MC1R) and MC5R creates a ligand-dependent signal modulation: Potential participation in physiological color change in the flounder. Gen. Comp. Endocrinol. 2016,230–231: 103–109.
- Matsuda, N.; Kasagi, S.; Nakamaru, T.; Masuda, R.; Takahashi, A.; Tagawa, M. Left-right pigmentation pattern of Japanese flounder corresponds to expression levels of melanocortin receptors (MC1R and MC5R), but not to agouti signaling protein 1 (ASIP1) expression. Gen. Comp. Endocrinol. 2018, 262: 90–98.
- Liang, Y.; Grauvogl, M.; Meyer, A.; Kratochwil, C. F. Functional conservation and divergence of color-pattern-related agouti family genes in teleost fishes. J. Exp. Zoolog. B Mol. Dev. Evol. 2021, 336(5): 443–450.
- Section 2.2:While the initial steps of Illumina sequencing and quality control appear sound, a more comprehensive description of the analysis methods, biological context, and validation steps would enhance the rigor and clarity of the study. The authors provide information on the quality of sequencing data but it would be beneficial to describe the bioinformatics pipeline used for gene annotation and highlight any specific findings or differentially expressed genes between the control and PTU treatment groups. The significance of the study and the relevance of the transcriptome data should be clarified. What specific biological questions or hypotheses are these sequencing data meant to address? How does this fit into the broader context of research on erythrophore development and red coloration in fish? Did the authors perform any pilot study with assays to verify the expression of specific genes or pathways identified through transcriptome analysis?
Reply:
Thank you for your suggestion, the suggestion was adopted. We have provided more detailed information on key aspects of sequencing quality control, such as software, which has been modified in the revised manuscript. For methods to bioinformatics pipeline used for gene annotation, we have provided information in section 4.8 lines 475-479.
Lines 466-467
To obtain clean reads, raw reads were filtered using the high-throughput quality control (HTQC) package.
The significance of the transcriptome is twofold. Firstly, we aim to explore the physiological effects of PTU on fish. PTU as the most common melanin inhibitor, reports on PTU about fish are focused on phenology and physiological traits, but little know about its physiological side effects. It is necessary to explore these effects at the transcriptome level. Secondly, the red coloration-related genes were annotated and screened for qRT-PCR validation. We want to know what genes expressed during this period concerning carotenoid metabolism and pigment cells differentiation. Those gene may be critical for pigment cell differentiation and carotenoid metabolism. We have used simple sentences to make explanation in the revised manuscript.
Lines 170
To explore the physiological effects of PTU at the genetic level, six genes were selected for validation using qRT-PCR.
- Section 2.4: The approach to validate RNA-seq data with qRT-PCR appears sound, but providing more details on sample size, statistical analysis, and the biological context of gene selection would enhance the overall quality of the validation process. It's essential to mention the sample size (biological replicates) for both the control and PTU treatment groups in the qRT-PCR validation. Additionally, statistical analysis results (e.g., p-values) should be provided to assess the significance of the observed gene expression differences. The authors have chosen specific genes for validation based on their involvement in relevant pathways. However, it's helpful to briefly explain why these particular genes were selected and how their expression changes relate to the research's biological context. This would provide more insight into the significance of these findings. Including some methodological details, such as primer sequences used for qRT-PCR and the cycling conditions, can be valuable for readers who may want to replicate the experiments
Reply:
Thank you for your suggestion, the suggestion was adopted. In the qRT-PCR validation, we selected three biological replicates, and we have added information about biological replicates and P value. Methodological details include reaction and cycling conditions for qRT-PCR were provided in section 4.9 lines 497-500, and primer sequences used for qRT-PCR were provided in supplementary material table 1.
Lines 171
Each group included three biological replicates.
Lines 174-177
By qRT-PCR, the PTU treatment group showed higher mRNA expression levels of rlbp1b (P < 0.01), rbp1.1 (P < 0.01), soat2 (P < 0.01), apoa1 (P < 0.01), and rpe65a (P < 0.01) and lower levels of rh2 (P < 0.01) than the control group (Figure 4)
Lines 219-221
Expression levels of these genes were significantly higher at 36 hpf than at 24 and 72 hpf (P < 0.05), except for bco1 (P < 0.05)
In previous studies, PTU have influence in eye development, visual behavior, and retinoic acid signaling, but molecular regulation is unknow. We found genes related to these biologic process and pathways were altered in transcriptome. We chose these genes for validation to explain how PTU effect biologic process via specific genes. We have added explanation why these particular genes were selected.
Lines 253-255
In addition to inhibiting melanin synthesis, PTU alters substance transport and other metabolic pathways that indirectly influence melanogenesis.
- Section 2.5. There is a discrepancy in the sampling stages used in these two figures. Figure 1 documents erythrophore development at time points ranging from 12 to 60 hours after fertilization (hpf), while Figure 5 presents observations at 24, 36, and 72 hpf. The authors should address this inconsistency and provide a rationale for not using the same stages of development in both figures. It is essential to clarify whether the stages chosen for each figure were deliberate and why they were chosen as such. This will ensure the accuracy and reliability of the comparative analysis between the two figures.
Reply:
Thank you for your suggestion. In the revised manuscript, we have provided a rationale for not using the same stages of development in figure1 and figure 5, to ensure the accuracy and reliability of the comparative analysis between the two figures.
Lines 190-193
Considering the observation of erythrophore changed in spotted scat larvae of the PTU treatment group (Figure 1), larvae at 24, 36, and 72 hpf were chosen for the subsequent characteristics, pigments, and gene expression analysis for red coloration and erythrophore development.
Reviewer 2 Report
The paper identifies pigments and describe gene expression analysis during etrythropnhore development. The red pigments studied are of commercial significance, and the study is relevant to more species than the one studied, the spotted scat. I have only a few minor points that the authors might wish to consider.
Figure 1 provides little information. I believe a higher magnification would improve it, as it is hard to see the erythrophores the arrows are supposed to point at.
Figure 2 labelling should be expanded in order to make it self-explanatory. The same might apply to Figure 3.
Figure 5A labelling is actually confusing. Could more text be included in the pictures? The rest of Figure 5 as well as Figure 6 and 7 makes sense.
The text is otherwise adequate, my only critique applies to figure labelling and explanations.
Author Response
The paper identifies pigments and describe gene expression analysis during etrythropnhore development. The red pigments studied are of commercial significance, and the study is relevant to more species than the one studied, the spotted scat. I have only a few minor points that the authors might wish to consider.
Reply:
Thank you for your positive comments on our manuscript.
Figure 1 provides little information. I believe a higher magnification would improve it, as it is hard to see the erythrophores the arrows are supposed to point at.
Reply:
Thank you for your suggestion, the suggestion was adopted. We have provided Figure 1 with high resolution in the revised manuscript.
Figure 2 labelling should be expanded in order to make it self-explanatory. The same might apply to Figure 3.
Reply:
Thank you for your suggestion, your suggestion was adopted. We have expanded the labelling of Figure 2 and Figure 3 in the revised manuscript.
Lines 157-158
The circles represent gene ratio, and the colors represent P value, respectively.
Lines 183-184
The green and blue columns represent the control and PTU treatment groups, respectively.
Lines 186-188
The symbols “*” and “**” above the error bars indicate significant differences at the levels of P = 0.05 and P = 0.01, respectively.
Figure 5A labelling is actually confusing. Could more text be included in the pictures? The rest of Figure 5 as well as Figure 6 and 7 makes sense.
Reply:
Thank you for your suggestion, the suggestion was adopted. We have added more text in Figure 5A, 6 and 7 labelling to make it more understandable in the revised manuscript.
Lines 206-207
The high magnification of erythrophores in the head (b,e,h), the white boxed area and tail part (c, f, i), the red boxed area.
Lines 208-209
Red, green, and blue columns represent the data at 24, 36, and 72 h, respectively.
Lines 212-215
The red and yellow columns represent the concentration of carotenoids and pteridines, respectively. Data are presented as means ± SE. The symbols “*” and “**” above the error bars indicate significant differences at the levels of P = 0.05 and P = 0.01, respectively.
Lines 227
Red, green, and blue columns represent gene expression at 24, 36, and 72 h, respectively.
Lines 229-230
The symbols “*” and “**” above the error bars indicate significant differences at the levels of P = 0.05 and P = 0.01, respectively.
Lines 233-234
Genes related to carotenoid metabolism are highlighted by orange circles, and the genes involved in pteridine synthesis are marked by yellow circles.
The text is otherwise adequate, my only critique applies to figure labelling and explanations.
Reply:
Thank you for your suggestion. We have modified the figure labelling and explanations in the revised manuscript.
Reviewer 3 Report
Dear Editors,
The reviewed manuscript is well-structured, with clear objectives and a logical flow of information. The study provides valuable insights into the development of erythrophores and the associated gene expression patterns in spotted scat larvae. The obtained results may help improve the aquaculture production of ornamental fish species and contribute to our knowledge of pigment cell development in aquatic organisms. Because understanding the pigmentation processes in fish has potential implications in aquaculture, ecology, and evolutionary biology the reviewed study deserves to be published in the IJMS periodical. However, there are some areas where major improvements in clarity, quantitative detail, and significance could enhance the overall impact of the manuscript.
Detailed review can be found in the attached file.
Best regard

Dear Editors,
The reviewed manuscript is well-structured, with clear objectives and a logical flow of information. The study provides valuable insights into the development of erythrophores and the associated gene expression patterns in spotted scat larvae. The obtained results may help improve the aquaculture production of ornamental fish species and contribute to our knowledge of pigment cell development in aquatic organisms. Because understanding the pigmentation processes in fish has potential implications in aquaculture, ecology, and evolutionary biology the reviewed study deserves to be published in the IJMS periodical. However, there are some areas where major improvements in clarity, quantitative detail, and significance could enhance the overall impact of the manuscript.
Detailed review can be found in the attached file.
Best regard
Author Response
Dear Editors,
Dear Authors,
In the reviewed manuscript the authors investigated the development of erythrophores(red pigment cells) in spotted scat (Scatophagus argus) larvae by various analytical approaches. The research included microscopic observation of erythrophores (red pigment cells), identification and characteristics of pigments responsible for red coloration and transcriptomic analysis of genes responsible for red coloration in the species. For this purpose the authors compared larvae treated with PTU (which inhibits melanin formation) to control larvae in natural seawater.
The introduction of the manuscript provides a broad overview of the importance of fish body coloration and sets the stage for the specific research focus of the study, which is the development of erythrophores and the molecular mechanisms related to red coloration in spotted scat larvae. Suggestions for corrections, fixes, and updates that may enhance clarity and accuracy:
- In the paragraph discussing melanin and carotenoid pigments, it would be helpful to briefly define "endogenous" and "exogenously" synthesized pigments;
Reply:
Thank you for your suggestion, the suggestion was adopted. We have added definition of "endogenous" and "exogenously" synthesized pigments in the revised manuscript.
Lines 52-54
The complete biosynthetic pathway of melanin and pteridine pigments has been proposed, and these pigments are endogenous substances synthesized autonomously in organs and tissues of teleosts [11,12].
Lines 62-65
Carotenoids cannot be synthesized autonomously in the organs and tissues of animals but are derived from the diets in the external environment and stored in bodies. Carotenoids belong to exogenous pigments [20].
- When introducing genes associated with melanin regulation (e.g., mc1r and mc5r), consider providing the full gene names the first time they are mentioned and then use abbreviations in subsequent mentions;
Reply:
Thank you for your suggestion, the suggestion was adopted. We have added the full gene names (mc1r and mc5r) in the revised manuscript.
Lines 56-57
Mc1r (melanocortin 1 receptor) and Mc5r (melanocortin 5 receptor).
- In the paragraph introducing spotted scat, consider including a sentence or two about why this species is particularly interesting or relevant for studying coloration;
Reply:
Thank you for your suggestion, the suggestion was adopted. We have included sentences to introduce relevant of this species for studying coloration.
Lines 85-90
Red coloration is essential for ornamental value in fish, erythrophore development, and red pigmentation in the species remains to be clarified. In spotted scat, erythrophores are present at the embryonic stage even before hatching, making it an ideal species for studying erythrophore formation and red coloration. However, erythrophores in the skin are gradually covered by melanophores when they are formed.4. While the research objective is clear, you might consider briefly mentioning the specific techniques or methods (e.g., transcriptome analysis, LC-MS, qRT-PCR) that will be used in the study;
Reply:
Thank you for your suggestion, the suggestion was adopted. We have mentioned transcriptome technique in the introduction.
Lines 84-85
The transcriptome is an important technique for studying body color. Black spot formation and melanophore coloration have been evaluated[33].
- Consider adding a sentence or two about potential applications or implications of the research findings – how might the insights gained be applied in aquaculture or conservation efforts?
Reply:
Thank you for your suggestion, the suggestion was adopted. We have added a sentence about potential applications of our research findings.
Lines 98-100
The results will facilitate an understanding of the molecular mechanisms of erythrophore differentiation and red coloration in other marine fish species and may benefit the selective breeding programs for ornamental and cultured fish.
The Materials and Methods section provides a detailed description of the experimental design, procedures, and data analysis methods. Overall, the section is comprehensive and provides the necessary details for replicating the study. Following clarifications and minor additions, that can further enhance the clarity and completeness of the methodology description are suggested:
- Ensure consistent units when specifying concentrations;
Reply:
Thank you for your suggestion. We have corrected the concentration units.
- Specify the temperature conditions during the sonication process;
Reply:
Thank you for your suggestion. We have added the temperature of the sonication process.
- Clarify the purpose of using nitrogen blowing in the procedure and the reason for adding acetonitrile-saturated hexane;
Reply:
Thank you for your suggestion. We have added the purpose for using nitrogen blowing, and reason for adding acetonitrile-saturated hexane was provided.
Lines 397-398
In order to completely remove the residual liquid, the supernatants were transferred into a nitrogen blowing tube (10 mL) and dried under a stream of nitrogen.
Lines 400-401
To remove the ester, 3 mL of acetonitrile-saturated hexane (hexane: acetonitrile 8:1 (vol.:vol.)) was added.
- Explain the rationale for using a 0.22 μm membrane when filtering the supernatant.
Reply:
Thank you for your suggestion. We referred to the experimental methods from reference 56 and we have added the reference in the revised manuscript.
- Include a brief explanation of why MRM mode was used for confirmation;
Reply:
Thank you for your suggestion. We referred from the reference, and we added the reference in the revised manuscript.
Lines 411
Multiple reaction monitoring (MRM) mode was used to confirm the presence of analytes [81],
81.Arrizabalaga-Larrañaga, A.; Rodríguez, P.; Medina, M.; Santos, F. J.; Moyano, E. Simultaneous analysis of natural pigments and E-141i in olive oils by liquid chromatography-tandem mass spectrometry. Anal. Bioanal. Chem. 2019, 411(21), 5577–5591.
- Specify the mass spectrometry conditions, such as ionization mode and source parameters;
Reply:
Thank you for your suggestion. We have added ionization mode in the revised manuscript, and more parameters such as gas temperature (°C), gas flow (L/min) were provided in lines 414-418. The reference values for MRM ion pairs, fragmentation voltage, and collision energy are shown in Table 1.
Lines 413-414
The ionization mode was electrospray ionization (ESI), and the eluent was monitored by ESI in positive mode.
- Specify the software version used for data analysis (e.g., DESeq2 version 1.16.3);
Reply:
Thank you for your suggestion. We have added the software version in the revised manuscript.
R(version 3.6.3), GOseq R packages (version 1.24.0), KOBAS(version 3.0)
- Include the accession number or link to the reference genome.
Reply:
Thank you for your suggestion. The accession number and link to the reference genome were provided in lines 471-473.
Based on the spotted scat reference genome sequence (https://ngdc.cncb.ac.cn/search/?dbId=gwh&q=GWHAOSK00000000.1; accession number GWHAOSK00000000.1)
The Results chapter provides detailed findings from the experimental work. Overall, the Results chapter effectively presents the experimental findings, providing clear descriptions, supported by appropriate data and analyses. The information appears to be accurate and essential for addressing the research questions. There are a few minor suggestions for improvements:
- Maintain consistent verb tense throughout the chapter. For instance, if you're describing a method used in the past, use the past tense consistently;
Reply:
Thank you for your suggestion. We have checked the verb tense, and corrected in the revised manuscript.
- Review the text for any redundant or unnecessary sentences. Aim for concise and clear descriptions of your findings;
Reply:
Thank you for your suggestion. We have reviewed the text, and deleted unnecessary sentences in the revised manuscript.
- In some cases, subsections could potentially be merged for a smoother flow of information. For instance, you might combine the "Illumina Sequencing and Annotation" and "Functional Enrichment Analyses of DEGs" subsections into a single section discussing the sequencing and analysis process.
Reply:
Thank you for your suggestion. We have combined the "Illumina sequencing and annotation" and "Functional enrichment analyses of DEGs" into a single section.
Lines 120-121
2.2. Illumina sequencing and annotation, functional enrichment analyses of differentially expressed genes (DEGs)
The Discussion chapter provides an in-depth analysis of the research findings related to pigment formation and gene expression in spotted scat during embryonic development. In general, the Discussion chapter provides valuable insights into the research findings. Here are some specific proposals for corrections, fixes, and updates:
- Begin the Discussion section with a concise summary of the main objectives and hypotheses of the study. This will help readers understand the context and focus of the discussion;
Reply:
Thank you for your suggestion. We have added a concise summary of the main objectives of our work in discussion section.
Lines 236-237
Red coloration is considered an economically important trait in ornamental fish species. The mechanism underlying red coloration is not well-understood.
Lines 239-241
To characterize the development of erythrophore and red coloration in spotted scat, inhibitor treatment, transcriptome analysis, LC-MS, and spectrophotometry were used.
- When discussing the effects of PTU on melanin pigment formation and gene expression, consider acknowledging the potential limitations of PTU as a tool. For example, discuss whether PTU may have off-target effects or affect other pathways that indirectly influence pigment formation;
Reply:
Thank you for your suggestion. We have added discussion for potential limitations of PTU.
Lines 253-255
In addition to inhibiting melanin synthesis, PTU alters substance transport and other metabolic pathways that indirectly influence melanogenesis.
- For genes whose expression is significantly altered by PTU treatment, provide a clearer link between these changes and the observed phenotypic outcomes.
Reply:
Thank you for your suggestion. In this experiment, PTU with 0.003% was used as a common and safe concentration for embryo observations such as zebrafish, reportedly has minimal other effects. This is why we chosen 0.003% for this experiment. We did not observe significant phenotypic outcomes except melanin was inhibited, or other outcomes are difficult to observe. Another possible reason, treatment time was too short, not enough to alter the obvious phenotypic outcomes, although the gene expressions have changed.
- In the discussion of retinol metabolism, elaborate on how the alteration of this pathway might affect the overall development of spotted scat embryos. Connect the potential impact on retinol metabolism to broader developmental processes beyond coloration;
Reply:
Thank you for your suggestion. We have added discussion about the possible affects altered by retinol metabolism.
Lines 274-276
In zebrafish, the knockout of rlbp1b impairs retinol metabolism, causing subretinal lipid deposits and photoreceptor degeneration [50].
Lines 277-279
rbp1 is regarded as an intracellular regulator of vitamin A metabolism and retinol transport, altering African clawed frog (Xenopus laevis) anterior neural development [52].
Lines 281-283
PTU may alter lipid metabolism, photoreceptor development, and neural development via retinol metabolism in spot scat, but these phenotypes require further study.
- Carefully proofread the text for language and grammar to ensure clarity and readability. Simplify complex sentences for easier comprehension where necessary.
Reply:
Thank you for your suggestion. We have proofread the text for language and grammar, and simplified sentences in the revised manuscript.
The conclusion chapter provides a concise summary of the key findings and insights obtained from your study. Following clarifications and minor additions, that can further enhance the clarity and completeness of the methodology description are suggested
- The conclusion should begin by briefly summarizing the main findings of the study;
Reply:
Thank you for your suggestion. We have added briefly summaries about main findings of our study.
Lines 508-509
In this study, we characterized the erythrophore development, identified the pigments, and analyzed the expression of genes responsible for red coloration in spotted scat.
- Be more specific about the findings related to erythrophore development, red coloration, and gene expression;
Reply:
Thank you for your suggestion. We have added more details about our findings in the revised manuscript.
Lines 511-512
CD of erythrophores was clearly visible from 24 hpf, and the number and diameter of cells showed an initial increase (from 14-36 hpf), followed by a decrease (from 48-72 hpf).
- Highlight any novel insights or contributions that your study has made to the field of chromatophore differentiation and coloration in aquatic animals;
Reply:
Thank you for your suggestion. We have added contributions about this study for aquaculture industry.
Lines 519-521
These findings provide valuable insights into the regulation of red coloration in aquatic animals and benefit the breeding programs of ornamental fish species.
- Suggest potential avenues for future research based on the gaps or unanswered questions that your study has identified. What aspects of erythrophore development or pigment regulation require further investigation?
Reply:
Thank you for your suggestion. We have proposed questions that may require further investigation.
Lines 521-522
The functions, regulation of erythrophore differentiation, and cell fate of these genes need to be studied and clarified in more fish species.
The abstract is generally clear and well-structured. It begins with a brief introduction, followed by the description of methods, results, and conclusions. The use of PTU to inhibit melanin formation is mentioned, but it might be helpful to provide a bit more detail about this method in the abstract. For instance, a brief explanation of how PTU works to inhibit melanin formation could enhance understanding. While the abstract mentions that the findings provide insights into chromatophore differentiation and gene function, it could be strengthened by briefly highlighting the broader significance of these insights. How might this research impact aquaculture or our understanding of coloration in aquatic animals?
Reply:
Thank you for your suggestion. We have made an explanation of how PTU works to inhibit melanin formation, and highlighted the significance of these insights.
Lines 18-19
1-phenyl 2-thiourea (PTU) is a tyrosinase inhibitor commonly used to inhibit melanogenesis and contribute to the visualization of embryonic development.
Lines 33-35
These findings provide insights into pigment cell differentiation and gene function in the regulation of red coloration and contribute to selective breeding programs for ornamental aquatic animals.
In general, the manuscript is well-structured, with clear objectives and a logical flow of information. The study provides valuable insights into the development of erythrophores and the associated gene expression patterns in spotted scat larvae. The obtained results may help improve the aquaculture production of ornamental fish species and contribute to our knowledge of pigment cell development in aquatic organisms. Because understanding the pigmentation processes in fish has potential implications in aquaculture, ecology, and evolutionary biology the reviewed study deserves to be published in the IJMS periodical. However, there are some areas where major improvements in clarity, quantitative detail, and significance could enhance the overall impact of the manuscript.
Reply:
Thank you for your positive comments on our work. We will try our best to revise the manuscript
Round 2
Reviewer 1 Report
Thank you addressing my comments and suggestions.
Reviewer 3 Report
Dear Editors,
Dear Authors,
The manuscript entitled: “Pigment identification and gene expression analysis during erythrophore development in spotted scat (Scatophagus argus) larvae” has been significantly improved. The Authors have regarded all my remarks and suggestions. The manuscript represents very good quality study. I do not have any other remarks and the manuscript can be published in the present form. Nice work!
Best regards,